Genome-wide association study and population structure analysis of seed-bound amino acids and total protein in watermelon

http://orcid.org/0000-0001-6803-4379 Joshi Vijay 1 2 vijay.joshi@tamu.edu
Nimmakayala Padma 3
http://orcid.org/0000-0002-9402-0090 Song Qiushuo 1
Abburi Venkata 3
http://orcid.org/0000-0001-5505-7109 Natarajan Purushothaman 3
Levi Amnon 4
Crosby Kevin 1
http://orcid.org/0000-0003-4611-7143 Reddy Umesh K. 3 ureddy@wvstateu.edu
1 Department of Horticultural Sciences, Texas A&M University , Uvalde, Texas , United States
2 Texas A&M AgriLife Research and Extension Center , Uvalde, Texas , United States
3 Department of Biology, Gus R. Douglass Institute, West Virginia State University, Institute , Charleston, West Virginia , United States
4 Vegetable Laboratory, USDA-ARS , Charleston, South Carolina , United States
Xu Xiangyang
Electronic publication date: 2021 Oct 19
Publication date: 2021
Volume: 9
Electronic Location ID: e12343
Received 2021 Apr 30; Accepted 2021 Sep 28
Copyright: © 2021 Joshi et al.
Copyright year: 2021
Copyright holder: Joshi et al.
License: This is an open access article distributed under the terms of the Creative Commons Attribution License, which permits unrestricted use, distribution, reproduction and adaptation in any medium and for any purpose provided that it is properly attributed. For attribution, the original author(s), title, publication source (PeerJ) and either DOI or URL of the article must be cited.
License URL: https://creativecommons.org/licenses/by/4.0/

Keywords: Watermelon, GWAS, Amino acids, Proteins

Funding: USDA-NIFA 2017-38821-26434 Hatch project #TEX09647 This work was supported by USDA-NIFA 2017-38821-26434 and Hatch project (#TEX09647). The funders had no role in study design, data collection and analysis, decision to publish, or preparation of the manuscript.

==============================
Background

Watermelon seeds are a powerhouse of value-added traits such as proteins, free amino acids, vitamins, and essential minerals, offering a paleo-friendly dietary option. Despite the availability of substantial genetic variation, there is no sufficient information on the natural variation in seed-bound amino acids or proteins across the watermelon germplasm. This study aimed to analyze the natural variation in watermelon seed amino acids and total protein and explore underpinning genetic loci by genome-wide association study (GWAS).

Methods

The study evaluated the distribution of seed-bound free amino acids and total protein in 211 watermelon accessions of Citrullus spp, including 154 of Citrullus lanatus, 54 of Citrullus mucosospermus (egusi) and three of Citrullus amarus. We used the GWAS approach to associate seed phenotypes with 11,456 single nucleotide polymorphisms (SNPs) generated by genotyping-by-sequencing (GBS).

Results

Our results demonstrate a significant natural variation in different free amino acids and total protein content across accessions and geographic regions. The accessions with high protein content and proportion of essential amino acids warrant its use for value-added benefits in the food and feed industries via biofortification. The GWAS analysis identified 188 SNPs coinciding with 167 candidate genes associated with watermelon seed-bound amino acids and total protein. Clustering of SNPs associated with individual amino acids found by principal component analysis was independent of the speciation or cultivar groups and was not selected during the domestication of sweet watermelon. The identified candidate genes were involved in metabolic pathways associated with amino acid metabolism, such as Argininosuccinate synthase, explaining 7% of the variation in arginine content, which validate their functional relevance and potential for marker-assisted analysis selection. This study provides a platform for exploring potential gene loci involved in seed-bound amino acids metabolism, useful in genetic analysis and development of watermelon varieties with superior seed nutritional values.

Introduction

Nutritional improvement of crop plants is most critical for the overall health of people around the world. Proteins and their structural constituents, amino acids, are indispensable for human nutrition and health. Plant seeds store a wide range of compounds, such as proteins, free amino acids, carbohydrates, and storage lipids, contributing ∼70% of the world’s human caloric intake directly or indirectly as animal feed (Sreenivasulu & Wobus, 2013). Furthermore, amino acids play vital roles in the central metabolism of seeds and are used to synthesize seed storage proteins as precursors for the biosynthesis of secondary metabolites and are catabolized via the tricarboxylic acid cycle to generate energy to support seedling growth (Amir, Galili & Cohen, 2018; Angelovici et al., 2011; Galili et al., 2014).

Humans and farm animals cannot synthesize many essential amino acids. Populations in low-income countries that rely on a few selected crops with an imbalanced amino acid composition develop health problems due to essential amino acid deficiencies. Manufactured animal feed is augmented with chemically synthesized amino acids, which is cost-intensive. Hence, strategies to increase protein levels and the concentration of essential amino acids in food crops are of primary importance in most crop improvement programs.

Several traditional and transgenic approaches have been successfully used to improve protein contents and amino acid balance in plant seeds (Chakraborty et al., 2010; Jiang et al., 2016; Newell-McGloughlin, 2008).

Watermelon seeds have been used as a principal staple food for native people in western Africa and the Sahara Desert (Jensen et al., 2011). The watermelon seeds are considered a potential powerhouse of proteins (30–35%), oils, B vitamins, niacin, thiamin, and essential minerals such as iron, magnesium, manganese phosphorus, potassium, and zinc. Because of the high protein content, omega-3 and-6 fatty acids, micronutrients, and lack of oligosaccharides (leads to flatulence in bean-based diets) (El-Adawy & Taha, 2001; Jyothi Lakshmi & Kaul, 2011; Rakhimov, Érmatov & Aliev, 1995; Wani et al., 2011b) and various essential amino acids and non-protein amino acids (El-Adawy & Taha, 2001; Jyothi Lakshmi & Kaul, 2011), watermelon seeds offer a paleo-friendly and gluten-free healthy dietary option. Arginine is one of the predominant amino acids in watermelon seeds (Hartman et al., 2019; Perkins-Veazie et al., 2015).

Genetic screens with various approaches such as linkage mapping, quantitative trait loci, and genome-wide association study (GWAS) have been used to genetically improve the accumulation of nutritionally limiting seed-bound amino acids or proteins in soybean (Lee et al., 2019; Panthee et al., 2006; Warrington et al., 2015; Zhang et al., 2018), chickpea (Upadhyaya et al., 2016), maize (Deng et al., 2017) and Arabidopsis (Angelovici et al., 2013; Jander et al., 2004; Joshi et al., 2006). Despite the availability of substantial genetic variation, the genes regulating amino acids or protein content in watermelon seeds and the extent of variability for seed amino acids and total protein across watermelon germplasm have not been studied. In this study, genetic components underlying natural variation and amino acid metabolism in watermelon seeds have been elucidated. We evaluated total amino acid and protein content in seeds of accessions representing different Citrullus spp. of the germplasm collection available at the USDA germplasm repository. We conducted GWAS and identified single nucleotide polymorphism (SNP) markers associated with watermelon seed-specific amino acids and total protein. The candidate genes identified in this study provide insights into amino acid and protein biosynthesis and could be used in marker-assisted selection to improve the nutritional value of watermelon and cucurbit crop seeds.

Materials & methods

Seed sample preparation

Seeds of 154 Citrullus lanatus, 54 Citrullus mucosospermus, and three Citrullus amarus accessions obtained from the USDA Germplasm Resources Information Network were evaluated for amino acid content and total protein. The seed coats were removed by using pliers or nail clippers to recover the intact endosperms. Approximately 20 mg and 10 mg shelled watermelon seeds of each accession were placed in a two-mL microcentrifuge tube in triplicate for amino acid and protein extraction. The samples were flash-frozen in liquid nitrogen and homogenized to a fine powder using 5 mm Demag stainless steel balls (Abbott Ball Co., West Hartford, CT, USA) in a Harbil model 5G-HD paint shaker.

Amino acid extraction and analysis

Amino acids were extracted using an established protocol (Joshi et al., 2019) by suspending the homogenized samples in 100 mM cold HCl extraction buffer, then incubation on ice (~20 min) and centrifugation @14,609×g for 20 min at 4 °C. The supernatants were collected and filtered through a 96-well 0.45-μm-pore filter plate (Pall Life Sciences, New York, USA). The eluents collected in 96-well trap plates were stored at −20 °C for further amino acid quantification. The derivatization of filtrates was carried out using the AccQ•Tag™ 3X Ultra-Fluor derivatization kit (Waters Corp., Milford, MA, USA) following the standard protocol. L-Norvaline (Sigma, St. Louis, MO, USA) was used as an internal standard. Amino acid calibration was performed using the Kairos™ Amino Acid Kit (Waters Corp., Milford, MA, USA).

Calibration curves were built with the TargetLynx™ Application Manager (Waters Corp., Milford, MA, USA). Amino acid detection was carried out using a Waters Acquity H-class UPLC system equipped with Waters Xevo TQ mass spectrometer with an electrospray ionization (ESI) probe. The Waters Acquity H-class UPLC system consists of an autosampler, a binary solvent manager, a column heater, and a Water’s AccQ•Tag Ultra column (2.1 mm i.d. × 140 mm, 1.7-μm particles). The mobile phase consisted of a water phase (A) (0.1% formic acid v/v) and acetonitrile (B) (0.1% formic acid v/v) with a stable flow rate at 0.5 mL/min and column temperature set at 55 °C. The gradient of non-linear separation was as follows: 0–1 min (99% A), 3.2 min (87.0% A), 8 min (86.5% A), and 9 min (5% A). Finally, two μl of the derivatized sample was injected into the column for analysis. IntelliStart software (Waters Corp., Milford, MA, USA) was used to optimize each amino acid Multiple Reaction Monitoring (MRM) transition, collision energy values, and cone voltage. ESI source was set to 150 °C with gas desolvation flow rate 1,000 L/h, gas flow cone 20 L/h, desolvation temperature 500 °C, capillary voltage 2.0 kV, gas collision energy 15 to 30 V, and cone voltage 30 V for detecting all amino acids. Water’s MassLynx™ software was used for instrument monitoring and data acquisition. The TargetLynx™ Application Manager (Waters Corp., Milford, MA, USA) was used for data integration, calibration curves, and amino acid quantification.

Total protein extraction and analysis

Total protein was extracted from homogenized seeds samples of each accession using an extraction buffer (70 mM Tris HCl, 25 mM KCl, 1 mM MgCl2, five mM EDTA, 5% glycerol, 0.1% Triton X-100, and 15 mM β-mercaptoethanol). The extracts were filtered using a 96-well 0.45-μm-pore filter plate (Pall Life Sciences, New York, USA), and filtrates were used to determine protein content with the Bradford Protein Assay Kit (AMRESCO Inc., Solon, OH). Extracts of 20 µl were incubated with 180 µl buffer for 2 min in a 96-well microplate (F-bottom, Greiner Bio-One, Kremsmünster, Austria) before measuring the absorbance at 595 nm in a spectrophotometer (Multiskan GO, Thermo Scientific, Waltham, MA, USA). Before measuring samples, 0.5 mg/mL BSA solution was used to prepare a standard curve to detect protein concentrations, and the total protein was reported as µg per mg of seed.

Statistical procedures such as ANOVA, Student t-Test, and the Principal component analysis (PCA) were performed using JMPR 15.2.0 (SAS Institute Inc., Cary, NC, USA) statistical package.

Association analysis

For GWAS, the population structure Q matrix was replaced by the PC matrix. The PC matrix and identity by descent (IBD) were calculated by using the EIGENSTRAT algorithm (Patterson, Price & Reich, 2006) with the SNP & Variation Suite (SVS v8.8.1; Golden Helix, Inc., Bozeman, MT, USA) in SVS v8.1.5. GWAS involved a multiple-locus mixed linear model developed by the EMMAX method and implemented in SVS v8.1.5. We used a PC matrix (first two vectors) and the IBD matrix to correct population stratification. Manhattan plots for associated SNPs were visualized in GenomeBrowse v1.0 (Golden Helix, Inc). The SNP p-values from GWAS underwent false discovery rate (FDR) analysis. The details of 11,456 SNPs generated by genotype by sequencing (Nimmakayala et al., 2014; Wu et al., 2019) used for association analysis and resolving population structure are available as supplemental data with the cited papers.

Population structure with associated SNPs

To analyze the impact of amino acid accumulation in the global collection of cultivars and wild types on population structure, we generated the principal components, or eigenvectors, by principal component analysis (PCA) and corresponding eigenvalues were estimated by the associated SNPs.

Results

Phenotypic variation in the seed-bound free amino acids and total protein

We investigated the distribution of seed amino acids in 211 watermelon accessions, including 154 of Citrullus lanatus, 54 of Citrullus mucosospermus (egusi), and three of Citrullus amarus. We detected 28 amino acids, including 20 protein-bound and eight non-protein amino acids (such as GABA, citrulline, ornithine) in the seeds of all watermelon accessions. Nitrogen-rich amino acids such as glutamic acid (29.6%), followed by arginine (17.9%), aspartic acid (9.7%), and alanine (7.6%) were the most abundant amino acids in watermelon seeds (Fig. 1). ANOVA confirmed a significant variation in the seed-bound free amino acids and total protein (p < 0.05). Amino acids and the total protein content of seeds are influenced by genetic backgrounds, geographic origin, species, agronomic conditions, and postharvest processing. The percent distribution of all seed-bound amino acids is presented in Table S1. The highest proportion of arginine (40.6%) and citrulline (18.1%) in seeds was measured in accessions PI 470246 (C. lanatus) and PI 254740 (C. mucosospermus), respectively. The details of geographical origin and continents are in Table S2. We found significant differences in the proportion of the most abundant amino acids (glutamic acid, arginine, aspartic acid, and alanine) across continents and species (Figs. S1–S4). For example, glutamate content was significantly lower in European than North American and Asian accessions. Likewise, African accessions showed the lowest arginine content compared with European and North American accessions. Aspartic acid and alanine content were highest in African and South American accessions, respectively. Similarly, glutamate and arginine contents were high in C. mucosospermus, whereas C. lanatus accessions had high aspartic acid and alanine contents. The percentages of free amino acids were submitted to the principal component analysis (PCA), which revealed a clear separation of C. mucosospermus and the African continent. The first two principal components accounted for 35.6% of the total variance in the data, of which PC1 explained 23.3% of this variance and PC2 explained 12.2% (Fig. S5). The amino acids such as branched-chain amino acids (Isoleucine, Leucine, valine), proline, asparagine, and alanine contributed positively to the construction of PC1. In contrast, glutamate, arginine, aspartate citrulline contributed negatively to construct PC1. Furthermore, glutamate, aspartate, and argininosuccinic acid contributed positively to construct PC2, whereas arginine, histidine, citrulline, and ornithine contributed negatively.

Figure 1 Percent distribution of free amino acids across seeds of selected watermelon accessions.

The percentage protein distribution across accessions is in Fig. 2. The accession PI 172799 had the highest seed protein content (19.5%). The crude seed protein content in the commercial cultivars (Black Diamond, Charleston Gray, and Crimson Sweet) has ranged from 16% to 17.7% (Tabiri, 2016). The mean protein content across accessions by continent is provided in Table S3. The mean protein content in accessions was significantly higher in Europe (11.5%) and Africa (11.3%) than Asia (10.6%) and North America (10.5%). The distribution of protein content in C. lanatus, C. mucosospermus, and C. amarus is summarized in Fig. S6. The mean protein content was comparable in C. mucosospermus (11.4%) and C. lanatus (11.0%) and C. amarus (10.3%).

Figure 2 Percent variation in the seed protein content of watermelon accessions.

The accession details of the associated seed IDs and mean total protein content data is available in Table S3.

Overall population structure

We used 11,456 SNPs generated by genotype by sequencing (Nimmakayala et al., 2014; Wu et al., 2019) for resolving population structure. Principal component analysis (PCA) separated ancestral species C. mucasospermous (egusi), C. lanatus (wild, landrace, and cultivars) into two groups. The C. lanatus group showed an admixture of wild, landrace (semi-wild), and cultivars (Fig. 3A). Many cultivars formed as a single cluster. This PCA was superimposed with arginine content to understand clustering across accessions during breeding histories for arginine content. Egusi types, wild lanatus, landraces, and cultivars possessed low, medium, and high arginine content in all taxa in the study (Fig. 3B). Hence, we did not notice any selection for arginine during the domestication of sweet watermelon.

Figure 3 (A) Principal component analysis (PCA) separating ancestral species C mucasospermous (egusi), C lanatus (wild, landrace, and cultivars). (B) PCA superimposed with the arginine contents across accessions.

GWAS for various amino acid and total protein content

Our GWAS associated 4, 22, 11, 19, 6, 12, 6, 30, 3, 10, 5, 14, 12, 13, 8, 3, 5, 8, 7, 11, 5, 4, 6, 11, 14, 5, 8 and 7 SNPs with histidine, arginine, asparagine, glutamine, serine, glutamic acid, aspartic acid, citrulline, threonine, glycine, alanine, GABA, proline, L-ornithine, cystine, lysine, tyrosine, methionine, valine, isoleucine, leucine, phenylalanine, ethanolamine, hydroxylysine, alpha aminoadipic acid, kynurenine, tryptophan, and arginine succinic acid, respectively. We present association statistics indicating chromosomal location, significance, FDR, regression beta (positive or negative), the standard deviation of regression beta, sample size, call rate, phenotypic variance explained by the SNP, minor allele frequencies of associated SNPs, type of mutation, gene name, gene region, and harboring these SNPs for all detected amino acids in Table S4 and total protein in Table S5. Manhattan plots showing chromosome distribution of associations for 24 seed-bound free amino acids and total protein are in Figs. 4–12, respectively. Quantile–quantile (Q–Q) plots for various amino acids and total proteins are in shown Figs. S7 and S8, respectively.

Figure 4 Manhattan plots of genome-wide association analyses for seed-bound Histidine, Arginine, and Asparagine using mixed linear model (MLM).

Coordinates of 11 chromosomes are displayed along the X-axis as color blocks with the negative log 10 of the association p-value for each single nucleotide polymorphism displayed on the Y-axis.

Figure 5 Manhattan plots of genome-wide association analyses for seed-bound Glutamine, Serine, and Glutamic Acid using mixed linear model (MLM).

Coordinates of 11 chromosomes are displayed along the X-axis as color blocks with the negative log 10 of the association p-value for each single nucleotide polymorphism displayed on the Y-axis.

Figure 6 Manhattan plots of genome-wide association analyses for seed-bound Aspartic Acid, Citrulline, and Threonine using mixed linear model (MLM).

Coordinates of 11 chromosomes are displayed along the X-axis as color blocks with the negative log 10 of the association p-value for each single nucleotide polymorphism displayed on the Y-axis.

Figure 7 Manhattan plots of genome-wide association analyses for seed-bound Glycine, Alanine, and GABA using mixed linear model (MLM).

Coordinates of 11 chromosomes are displayed along the X-axis as color blocks with the negative log 10 of the association p-value for each single nucleotide polymorphism displayed on the Y-axis

Figure 8 Manhattan plots of genome-wide association analyses for seed-bound Proline, L-Ornithine, and Cystine using mixed linear model (MLM).

Coordinates of 11 chromosomes are displayed along the X-axis as color blocks with the negative log 10 of the association p-value for each single nucleotide polymorphism displayed on the Y-axis

Figure 9 Manhattan plots of genome-wide association analyses for seed-bound Lysine, Tyrosine, and Methionine using mixed linear model (MLM).

Coordinates of 11 chromosomes are displayed along the X-axis as color blocks with the negative log 10 of the association p-value for each single nucleotide polymorphism displayed on the Y-axis

Figure 10 Manhattan plots of genome-wide association analyses for seed-bound Valine, Isoleucine and Leucine using mixed linear model (MLM).

Coordinates of 11 chromosomes are displayed along the X-axis as color blocks with the negative log 10 of the association p-value for each single nucleotide polymorphism displayed on the Y-axis

Figure 11 Manhattan plots of genome-wide association analyses for seed-bound Phenylalanine, Tryptophan, and Argininosuccinic Acid using mixed linear model (MLM).

Coordinates of 11 chromosomes are displayed along the X-axis as color blocks with the negative log 10 of the association p-value for each single nucleotide polymorphism displayed on the Y-axis

Figure 12 Manhattan plots of genome-wide association analyses for total seed proteins using mixed linear model (MLM).

Coordinates of 11 chromosomes are displayed along the X-axis as color blocks with the negative log 10 of the association p-value for each single nucleotide polymorphism displayed on the Y-axis

Population structure analysis based on associated SNPs for amino acids and total protein

We used PCA with associated SNPs for each amino acid and total protein content in the study. The PCA revealed how associated SNPs representing causative genes distort the population structure compared with the overall population structure. The first two eigenvectors of PCA cumulatively explained the percentage variation absorbed by each amino acid (Fig. S9) and total protein (Fig. S10). Our study revealed associated SNPs when used for PCA analysis, individual PCA for histidine, arginine, asparagine, glutamine, serine, glutamic acid, aspartic acid, citrulline, threonine, glycine, alanine, GABA, proline, L-ornithine, cystine, lysine, tyrosine, methionine, valine, isoleucine, leucine, phenylalanine, ethanolamine, hydroxylysine, alpha aminoadipic acid, kynurenine, tryptophan, and arginine succinic acid absorbed 57%, 45%, 32%, 59%, 47%, 26%, 50%, 21%, 85%, 26%, 35%, 34%, 36%, 29%, 35%, 26%, 60%, 29%, 30%, 51%, 22%, 57%, 51% and 52%, respectively of total genetic variance. PCA with associated SNPs for serine, aspartic acid, threonine, lysine, phenylalanine, kynurenine, tryptophan, and arginine succinic acid divided the population into three distinct clusters, although the clustering was independent of the speciation or cultivar groups. None of the seed-bound amino acid components was selected during the domestication of sweet watermelon. Because the domestication of sweet watermelon is based on fruit size, rind thickness, flesh softening, and soluble solids, seed composition may not have been directly or indirectly under selection.

Genes under association

Detailed annotation for the genes identified for various metabolites in the current GWAS is in Table S4. Among the genes identified by GWAS that were common to various metabolites, the gene most directly and highly associated with various metabolites was a nonsynonymous SNP (S2_9832702) located in ATP-binding cassette (ABC) transporter B family member 19 (ABCB19; ClCG02G007990), located on chromosome 2 and showing multiple significant associations with alpha aminoadipic acid (p = 0.00000004), glutamine (p = 0.0002), glycine (p = 0.000004), GABA (p = 0.000004), proline (p = 0.0002), lysine (p = 0.000027), valine (p = 0.000006), leucine (0.000003), isoleucine (p = 0.000000006), ethanolamine (p = 0.0002), cysteine (p = 0.000005) and alpha aminoadipic acid (p = 0.00000004), which explained 9%, 10%, 8%, 11%, 7%, 9%, 11%, 9%, 8%, 5%, 7% and 11% of the phenotypic variance, respectively. This gene is known for mediating the polar transport of auxin and is required to establish an auxin concentration gradient, which is essential for cytoplasmic streaming that may have a role in seed development.

Argininosuccinate synthase (ClCG03G003660), located on chromosome 3 (p = 0.0008 FDR 0.003), explained 7% of the variance for both arginine and argininosuccinic acid. Arogenate dehydrogenase (TyrA/ADH; ClCG11G003430), on chromosome 11, was strongly associated with arginine (p = 0.00009), tyrosine (p = 0.0005) and argininosuccinic acid (p = 0.0003). S7_7819769, an exonic located in sspartate semialdehyde dehydrogenase (ClCG07G005480), was associated with asparagine (p = 0.0005), explaining 7% of the variance.

A haplotype with two SNPs in aspartate/tyrosine/aromatic aminotransferase (ClCG11G013620) was associated with alanine (p = 0.00002), explaining 9% of the variance; tyrosine (p = 0.0007), 7%; and asparagine (p = 0.0000000001), 10% of the variance. Tyrosine aminotransferase was significantly associated with arginine, alpha aminoadipic acid, glutamic acid, asparagine, alanine, L-ornithine, and tyrosine.

A strong haplotype encompassing S2_1683683 to S2_1683687 with five SNPs is located in pentatricopeptide repeat protein 65 (ClCG02G001590), associated with glutamine (p = 0.0002), explaining 8% of the variance. This gene was also significantly associated with citrulline (p = 0.0001), proline (p = 0.001), GABA (p = 0.0001), lysine (p = 0.0009) and alpha aminoadipic acid (p = 0.0001).

Another haplotype on chromosome 11 located in glutamine--tRNA ligase (ClCG11G004350) has three SNPs associated with citrulline (p = 0.0004).

ABA-responsive element-binding factor 2 (ClCG08G016000) was associated with serine (p = 0.001), explaining 7% of the variance; glycine (p = 0.0001), 9%; proline (p = 0.0000001), 12%; and valine (p = 0.0007), 7%.

Acyl-[acyl-carrier-protein] desaturase was associated in the synthesis of citrulline (p = 0.0000001), explaining 13% of the variance; glycine (p = 0.0001), 5%; L-ornithine (p = 0.0005), 8%; lysine (p = 0.0004), 8%; and glutamic acid (p = 0.0008) 5%.

Different ankyrin-repeat family proteins were associated with aspartic acid (p = 1.29E−06), explaining 10% of the variance; arginine (p = 0.0004), 8%; and total protein (p = 5.23E−08), 8%. Arogenerate dehydrogenase was associated with arginine (p = 9.21E−05), explaining 3% of the variance; tyrosine (p = 0.0006), 7%; and argininosuccinic acid (p = 0.0003), 5%.

E3 ubiquitin-protein ligase UPL3 (ClCG02G011880) was associated with the synthesis of methionine (p = 1.93E−08), explaining 8% of the variance; valine (p = 0.0009), 7%; isoleucine (p = 3.29E−07), 8%; ethanolamine (p = 8.17E−05), 8%; and alpha aminoadipic acid (p = 0.0003), 8%. Endo-1, 3, 1, 4 beta-d-glucanase was associated with threonine (p = 0.0006), explaining 7% of the variance; L-ornithine (p = 0.0008), 7%; and tyrosine (p = 0.0008), 7%.

DNA-directed RNA polymerase was associated with arginine (p = 4.73E−05), explaining 6% of the variance; citrulline (p = 2.92E−06), 12%; asparagine (p = 6.36E−05), 5%; serine (p = 8.69E−0), 9%; glycine (p = 2.67E−07), 10%; tyrosine (p = 5.10E−06), 9%; and methionine (p = 0.0008), 7%.

E3 ubiquitin-protein ligase UPL3 was associated with methionine (p = 1.93E−08), explaining 8% of the variance; valine (p = 0.001), 7%; isoleucine (p = 3.29E−07), 7%; ethanolamine (p = 8.17E−05), 10%; and alpha aminoadipic acid (p = 0.0004), 8%.

GATA transcription factor (ClCG10G013320) was associated with arginine (p = 0.0007), explaining 7% of the variance; glutamine (p = 0.0005), 8%; and citrulline (p = 6.78E−06), 12%. Leucine-rich receptor-like protein kinase was associated with citrulline (p = 6.15E−07), explaining 14% of the variance; L-ornithine (p = 0.0003), 18%; and argininosuccinic acid (p = 3.96E−06), 8%.

Oleosin, the putative gene, was associated with asparagine (p = 0.0003), explaining 8% of the variance; glutamine (p = 0.0004), 8%; and L-ornithine (0.0006), 7%.

Zinc finger family protein was associated with glutamine (p = 0.0001), explaining 9% of the variance; glutamic acid (p = 2.24E−05), 4%; argininosuccinic acid (p = 0.001), 7%; arginine (p = 0.0001), 9%; aspargine (p = 1.18E−05), 5%; L-ornithine (p = 4.36E−05), 7%; citrulline (p = 6.42E−05), 10%; tryptophan (p = 5.31E−05), 10%; aspargine (p = 0.0006), 7%; alanine (p = 2.72E−05), 7%; total protein (p = 0.0004), 8%; and hydroxylysine (p = 1.45E−06), 8%.

Transmembrane protein adipocyte-associated-1 homolog was significantly associated with glycine (p = 0.0003), explaining 8% of the variance; alanine (p = 0.0009), 7%; and valine (p = 0.0003), 8%. TOM1-like protein 2 was significantly associated with asparagine (p = 4.67E−09), explaining 9% of the variance; glutamic acid (p = 0.0002) 9%; threonine (p = 0.0007), 7%; glycine (p = 6.30E−07), 6%; tyrosine (p = 0.0005), 5%; and valine (p = 3.23E−05), 10%.

Nucleotide diversity and Tajima’s D for various amino acids

We estimated nucleotide diversity (π and ϴ) and Tajima’s D using the associated SNPs for various metabolites estimated in the current study (Table 1). These parameters were separately estimated for wild and cultivated watermelon. None significantly differed between wild and cultivated varieties, which indicated that the genetic mechanisms underlying seed-bound amino acids might not undergo any selection during domestication.

Table 1 Estimation of nucleotide diversity (π and ϴ) and Tajima’s D in wild accessions and cultivars using the associated SNPs for various metabolites.

Compound	Wild accessions	Cultivars	
	PiPerBP	ThetaPerBP	TajimaD	PiPerBP	ThetaPerBP	TajimaD	
Histidine	0.44727	0.17781	2.796	0.42612	0.19064	2.42097	
Arginine	0.37416	0.17781	3.07763	0.34586	0.19064	2.36865	
Asparagine	0.39409	0.17781	2.99071	0.38203	0.19064	2.59433	
Glutamine	0.34258	0.17781	2.52647	0.28994	0.19064	1.4843	
Serine	0.39736	0.17781	2.59423	0.39127	0.19064	2.33932	
Glutamic acid	0.33648	0.17781	2.23581	0.33306	0.19064	1.96538	
Aspartic acid	0.41423	0.17781	2.79362	0.40269	0.19064	2.47245	
Citrulline	0.41116	0.17781	3.80497	0.38377	0.19064	3.05821	
Threonine	0.28987	0.17781	1.04887	0.29278	0.19064	0.94971	
Glycine	0.50441	0.17781	4.4193	0.49677	0.19064	4.06485	
Alanine	0.39035	0.17781	2.37438	0.40995	0.19064	2.42216	
GABA	0.4673	0.17781	4.20796	0.46442	0.19064	3.89117	
Proline	0.4213	0.17781	3.43089	0.39976	0.19064	2.88584	
L-Ornithine	0.39879	0.17781	3.1656	0.38207	0.19064	2.68343	
Cystine	0.31258	0.17802	1.72256	0.32379	0.19064	1.677	
Lysine	0.35466	0.17781	1.65535	0.34738	0.19064	1.45748	
Tyrosine	0.35377	0.17781	1.96577	0.33973	0.19064	1.64662	
Methionine	0.43189	0.17781	3.25328	0.42781	0.19064	2.98728	
Valine	0.42894	0.17781	3.10179	0.4346	0.19064	2.96847	
Isoleucine	0.42913	0.17781	3.47525	0.40277	0.19064	2.87541	
Leucine	0.43068	0.17781	2.82495	0.42798	0.19064	2.62141	
Phenylalanine	0.30904	0.17781	1.36162	0.31859	0.19064	1.3155	
Ethanolamine	0.24421	0.17781	0.78456	0.25513	0.19064	0.75187	
Hydroxylysine	0.33186	0.17781	2.13023	0.29377	0.19064	1.39787	
Alpha aminoadipic acid	0.25084	0.17781	1.06154	0.26251	0.19064	1.02139	
Kynurenine	0.37664	0.17781	2.22127	0.37595	0.19064	2.04666	
Tryptophan	0.38067	0.17781	2.5975	0.35361	0.19064	2.05263	
Argininosuccinic acid	0.40357	0.17781	2.78851	0.37942	0.19064	2.29704	

Discussion

The current study explored natural variation for various seed amino acids and provides a list of cultivars, landraces, and egusi lines with high amino acid content for potential use in breeding programs. Watermelon seed proteins and free amino acids are among the most tangible paleo-friendly, nitrogen-rich, gluten-free dietary options. Seeds of egusi cultivars are a significant part of the diet in many African countries (Giwa & Akanbi, 2020). Giwa & Akanbi (2020) reviewed various reports indicating egusi melon seed kernels are a good source of edible oil (31–59%), protein (19–37%), fiber (3–4%), and carbohydrate (8–20%). A coarse whitish meal of grounded seed kernel of egusi is used to make nutritious and pleasantly nutty-tasting (Oke, 1965). A detailed metabolic landscape of watermelon seeds using the diversity panel such as the current study would help identify genetic loci and parental lines to develop value-added seeded varieties to promote seeds for human consumption and culinary use (food additives) and livestock consumption.

Nutritionally superior seeds can be developed by identifying and deploying the gene(s)/QTLs associated with seed-specific nutraceuticals. Understanding genetic and molecular regulation of seed-specific proteins would help identify targets for developing protein-rich food options. GWAS and QTL mapping has been used to analyze seed-specific protein, oil, fatty acid, and amino acid content in various crops such as soybean (Hwang et al., 2014; Lee et al., 2019; Zhang et al., 2019), chickpea (Upadhyaya et al., 2016), peanut (Gangurde et al., 2020) rice (Chen et al., 2018), cotton (Yuan et al., 2018), and maize (Cook et al., 2011). Although it is hard to generalize the outcomes due to the species-level differences in the protein quality and composition, these studies have allowed the identification of SNPs and unique biomarkers linked to QTL for genome manipulation, germplasm enhancement, and the creation of high-density gene libraries.

This study performed GWAS of amino acids in watermelon seed to efficiently identify genetic loci influencing amino acid profiles in watermelon and underlying candidate genes and networks. We used significantly associated SNPs for these amino acids to explore the population structure and compared them with the genome-wide population structure to understand whether the genetic mechanisms underlying various amino acid metabolism had no role in the domestication of sweet watermelon. Furthermore, nucleotide diversities and Tajima’s D estimated for the SNPs underlying the composition of amino acids did not differ between wild and cultivated watermelon genotypes, so seed amino acid profiles were not involved in domestication.

Although GWAS identified genes with strong associations with more than one metabolite, we explored genes with known roles in regulating these metabolites. The most intriguing nonsynonymous SNP in ABCB19 (ClCG02G007990) was associated with multiple amino acids such as alpha aminoadipic acid, glutamine, GABA, proline, lysine, valine, leucine, isoleucine, ethanolamine, and alpha aminoadipic acid. The encoded protein is conserved across Cucurbits and shares more than 67% identity with several plant species such as cotton, Medicago, and chickpea. Members of the ABC transporter superfamily transport a wide range of molecules across various membrane types (Do, Martinoia & Lee, 2018) and play an essential role in seed development, seed germination, organ formation, and secondary growth (Hwang et al., 2016). Regulation of ABCB19 involves intricate coordinated cellular processes, including protein-protein interactions, vesicular trafficking, protein phosphorylation, ubiquitination, and stabilization of the transporter complexes (Titapiwatanakun & Murphy, 2009). Establishing an auxin concentration gradient is required, which is essential for cytoplasmic streaming (Peer, Jenness & Murphy, 2014).

The tyrosine aminotransferase (TAT, ClCG11G017780), associated with multiple amino acids, represents the entry point for various tyrosine degradation of natural products’ biosynthesis in the recycling of energy and nutrients in plants (Wang et al., 2019). The Arabidopsis ortholog of this gene is induced under conditions leading to oxidative stress and senescing leaves, developing seeds, and ABA treatment (Holländer-Czytko et al., 2005; Winter et al., 2007). Similarly, the orthologs of the basic leucine zipper transcription factor ABA-responsive element binding factor 2, with multiple associations with amino acids, is a positive regulator of glucose signal transduction and upregulates ABA-responsive gene expression (Cutler et al., 2010). Besides modulating various plant developmental processes, ABA regulates seed maturation, dormancy, and germination (Choi et al., 2000).

E3 ubiquitin-protein ligase UPL3 (ClCG02G011880) identified in this study encodes an ortholog of Arabidopsis (Downes et al., 2003) and Brassica miller (Miller et al., 2019) E3 ligase UPL3. Brassica UPL3 modulates seed size and lipid content and is exploited to increase yield. GATA transcription factor (ClCG10G013320) associated with arginine glutamine and citrulline suggests its involvement in regulating plant nitrogen metabolism. Several studies support GATA transcription factors’ involvement in regulating nitrogen metabolism and amino acid-related genes in fungi and plants (Fu & Marzluf, 1990; Hudson et al., 2011; Rastogi et al., 1997; Shin et al., 2017).

Argininosuccinate synthase or synthetase (ASS; EC 6.3.4.5) regulates the antepenultimate step in arginine synthesis that catalyzes argininosuccinate synthesis from citrulline and aspartate. The progressive downregulation of three argininosuccinate synthases (ASS-1, ASS-2, and ASS-3) has been demonstrated in developing flesh and rind tissues from watermelon to accumulate free citrulline (Guo et al., 2013; Joshi et al., 2019; Zhu et al., 2017). However, citrulline content was relatively lower than arginine in mature seeds, implying deregulation of citrulline catabolism in seeds. According to the distribution of the seed-specific free (Joshi et al., 2019; Perkins-Veazie et al., 2015) and protein-bound amino acids (Wani et al., 2011a) in watermelon, arginine is one of the most abundant amino acid residues in the storage proteins. Arginine has the highest nitrogen-to-carbon ratio and is suitable as a storage form of organic nitrogen in seeds. About 40% to 50% of the total nitrogen reserve in various plants is represented by arginine (Aninbon et al., 2017; Cortés-Giraldo et al., 2016; de Ruiter & Kollöffel, 1983; King & Gifford, 1997; Micallef & Shelp, 1989b). Arginine plays a crucial role in nitrogen distribution and recycling (Slocum, 2005) and affects the synthesis of storage proteins critical for germination. In pea seeds, argininosuccinate synthetase activity was significantly increased during the first weeks after anthesis, peaking at about 35 days and decreasing progressively until 54 days (de Ruiter & Kollöffel, 1983), which indicates its prominent role in arginine synthesis in seeds. Similarly, validation of 72% of arginine requirement in developing soybean cotyledons by in situ biosyntheses (Micallef & Shelp, 1989b) and recovery of labeled arginosuccinate (Micallef & Shelp, 1989a) further support the functional role of arginine biosynthetic enzymes in seeds.

Arogenate dehydrogenase (TyrA/ADH; ClCG11G003430) was associated with arginine. In plants, the amino acid tyrosine is synthesized from arogenate by an enzyme arogenate dehydrogenase (TyrA). TyrA activity has been demonstrated in several plants (Byng et al., 1981; Connelly & Conn, 1986; Gaines et al., 1982; Rippert et al., 2009). Tyrosine is an essential aromatic amino acid required to synthesize proteins and serves as a precursor of several secondary metabolites families, including tocochromanols (vitamin E) and plastoquinones isoquinoline alkaloids and non-protein amino acids (Tzin & Galili, 2010). In maize kernels, reduction in abundance or activity of arogenate dehydrogenase reduced the zein protein synthesis rate (Holding et al., 2010). The broader impact of TyrA arogenate dehydrogenases on plant growth and development and seed yield was validated in Arabidopsis (de Oliveira et al., 2019).

Another important gene from this GWAS is the aspartate semialdehyde dehydrogenase (ClCG07G005480), regulating the master aspartate pathway that leads to the synthesis of lysine methionine and threonine (Jander & Joshi, 2009). Although little is known about this gene’s functional significance in plants, its unique place in the aspartate master pathway is indispensable for synthesizing downstream limiting amino acids. Consistent with this concept, increased aspartate semialdehyde content in transgenic rice seeds overexpressing feedback-insensitive aspartate kinase (AK) was converted to homoserine and threonine in seeds (Long et al., 2013), which suggests the role of this gene as a gatekeeper for regulating the synthesis of threonine, methionine, or lysine.

Because the current GWAS involved relatively few SNPs, it is limited in identifying long haplotypes involving candidate genes. Nonetheless, this study identified a long haplotype in pentatricopeptide repeat protein 65 (PPR, ClCG02G001590) with a highly significant association with glutamine as well as four other amino acids. Mutant analysis in Arabidopsis validated that PPR proteins significantly impact seed development and plant growth (de Longevialle et al., 2007) and maize (Gutiérrez-Marcos et al., 2007; Manavski et al., 2012). Recently, a defective kernel maize mutant Dek53 encoding a PPR protein with lethal embryo and collapsed endosperm function was characterized in maize (Dai et al., 2020) confirms the PPR role in seed development.

In our GWAS analysis of total seed protein, the study identified a gene (ClCG03G016270) annotated as late embryogenesis abundant (LEA) protein. This protein’s amino acid sequence shares high homology (>90%) with other known LEA proteins in plants. LEA proteins accumulate to high levels during the last stage of seed formation (Hundertmark & Hincha, 2008) and play various roles, especially induction during water deficit (Tunnacliffe & Wise, 2007). LEA proteins also play a crucial role in normal seed development and plant growth (Manfre, Lanni & Jr Marcotte, 2006). Recently, the presence of the seed maturation protein domain and a progressive increase in the expression of this gene (ClLEA-25; Cla011279) during fruit maturation was validated in watermelon (Celik Altunoglu et al., 2017).

Conclusions

Watermelon seeds are a staple food for people in different parts of the world. Watermelon seed proteins and free amino acids contribute to the most tangible paleo-friendly protein-rich and gluten-free dietary options. Our results demonstrate a significant natural variation in different free amino acids and total protein content across accessions and geographic regions. The accessions with high protein content and a proportion of essential amino acids can be used for value-added benefits in the food and feed industries via biofortification. This study is the first to reveal the genetic architecture of seed-bound amino acids in a watermelon GWAS of 211 diverse accessions of Citrullus spp. with 11,456 SNPs generated by genotyping by sequencing (GBS) analysis. The GWAS identified quantitative gene loci (QTL) and several candidate genes involved in the metabolism of individual amino acids. The candidate genes identified here could help study the seed-bound amino acid accumulation, facilitate marker-assisted selection and provide novel targets for editing to accelerate nutrigenomics and associated breeding programs.

Supplemental Information

Supplemental Information 1 Contour graphs showing variation in the percent glutamic acid content in watermelon accessions across geographies and species.

The levels not connected by the same lower case (continents) and upper case (species) letters are significantly different (Student t-test; p = 0.05)

Click here for additional data file.

Supplemental Information 2 Contour graphs showing variation in the percent arginine content in watermelon accessions across geographies and species.

The levels not connected by the same lower case (continents) and upper case (species) letters are significantly different (Student t-test; p = 0.05)

Click here for additional data file.

Supplemental Information 3 Contour graphs showing variation in the percent aspartic acid content in watermelon accessions acrosss geographies and species.

The levels not connected by the same lower case (continents) and upper case (species) letters are significantly different (Student t-test; p = 0.05)

Click here for additional data file.

Supplemental Information 4 Contour graphs showing variation in the percent alanine content in watermelon accessions across geographies and species.

The levels not connected by the same lower case (continents) and upper case (species) letters are significantly different (Student t-test; p = 0.05)

Click here for additional data file.

Supplemental Information 5 Contour graphs showing geographic variation in protein content across continents (A) and species (B) in watermelon accessions.

Click here for additional data file.

Supplemental Information 6 Principal component analysis (PCA) of seed-bound amino acids.

Bi-plot for the first two components (PC) is shown for all seed-bound free amino acids in watermelon accessions (A) across continents (B) and species (C)

Click here for additional data file.

Supplemental Information 7 Q-Q (quantile-quantile) plots of different amino acids.

The plots were drawn for expected vs. observed −log10 (p-values) for all amino acids

Click here for additional data file.

Supplemental Information 8 Q-Q (quantile-quantile) plots of total proteins.

The plots was drawn for expected vs. observed −log10 (p-values) for total seed proteins

Click here for additional data file.

Supplemental Information 9 Principal component analysis (PCA) for amino acids showing the components of population genetic variation.

Click here for additional data file.

Supplemental Information 10 Principal component analysis (PCA) of total proteins showing the components of population genetic variation.

Click here for additional data file.

Supplemental Information 11 Mean percent distribution of seed amino acids in watermelon accessions.

Click here for additional data file.

Supplemental Information 12 Origin and distribution of the watermelon accesions.

Click here for additional data file.

Supplemental Information 13 Percent total protein distribution across watermelon accessions.

Click here for additional data file.

Supplemental Information 14 SNP chromosomal positions and candidate genes significantly associated with individual amino acids identified by GWAS.

Click here for additional data file.

Supplemental Information 15 SNP chromosomal positions and candidate genes significantly associated with total protein identified by GWAS.

Click here for additional data file.

Supplemental Information 16 Raw Data for Percent Seed Protein.

Click here for additional data file.

We appreciate Ms. Tatum Story for her assistance in sample preparation for metabolic analysis.

Additional Information and Declarations

Competing Interests

Author Contributions

Data Availability

The authors declare that they have no competing interests.

Vijay Joshi conceived and designed the experiments, prepared figures and/or tables, authored or reviewed drafts of the paper, and approved the final draft.

Padma Nimmakayala performed the experiments, prepared figures and/or tables, and approved the final draft.

Qiushuo Song performed the experiments, prepared figures and/or tables, and approved the final draft.

Venkata Abburi analyzed the data, prepared figures and/or tables, and approved the final draft.

Purushothaman Natarajan analyzed the data, prepared figures and/or tables, and approved the final draft.

Amnon Levi conceived and designed the experiments, authored or reviewed drafts of the paper, and approved the final draft.

Kevin Crosby conceived and designed the experiments, authored or reviewed drafts of the paper, and approved the final draft.

Umesh K. Reddy conceived and designed the experiments, prepared figures and/or tables, authored or reviewed drafts of the paper, and approved the final draft.

The following information was supplied regarding data availability:

The data is available in Table 1 and the Supplemental File.

The details of 11,456 SNPs generated by genotype by sequencing (Nimmakayala et al., 2014; Wu et al., 2019) used for association analysis and resolving population structure are available as supplemental data in the cited articles.

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
