# Peer review of "Genome-wide association study and population structure analysis of seed-bound amino acids and total protein in watermelon"

_PeerJ, doi:10.7717/peerj.12343_

## Round 0.1 · original submission · Major Revisions

Dear Authors,

The process of review was completed. Review reports have highlighted the interesting work and experimental design. In contrast, some part of the discussion and conclusion should be modified structurally and strengthened.

·

Basic reporting

In this manuscript, the authors used a nature watermelon panel for GWAS analysis to detected the loci with seed-bound amino acids and total protein. The content was clear with professional English. References related with the content were also provided. The structure of manuscript was good and clear. The figures in the review version seemed in low-resolution.

Experimental design

The experiment design was good. I suggest the author make more analysis about the SNPs which you have got in the GWAS results for a further verification.

Validity of the findings

Some SNPs associated with the amino acids and total protein content in watermelon seeds were detected. Genes located in the mapping region were also analyzed in the discussion part. The speculation in the conclusions should be verified more.

Additional comments

1. The authors used PCA results instead of population structure. For the GWAS panel, the population structure was not only PCA analysis. STRUCTURE and Admixture software would be used for population structure analysis.
2. The content from Line 255 to Line 314,was repetition of similar structures. As a suggestion, a table would be a better choice.
3. For the GWAS analysis, the Q-Q plot should be added.
4. According to the Manhattan plots, most traits exhibited some associated SNPs up the threshold line, this may due to the low re-sequencing depth. The seed amino acids and total protein seemed regulated by multiple genes as the quantitative trait . Due to the huge work for measurement these traits in multiple environments or years. I think the author should compare the SNP genotype (or the haplotype) with the phenotype to verify the accuracy of GWAS results. Maybe we can found some new information based on the above analysis. These SNP may only existed in some subspecies or variety.
5. For all the SNPs detected in the GWAS results? Is there some locus can used as the MAS tool ?
6. In the Results and Discussion part, the author listed some candidate genes in the associated region. Is there some sequences variations in the candidate genes between different accessions ?

Reviewer 2 ·

Basic reporting

no comment

Experimental design

no comment

Validity of the findings

no comment

Additional comments

This manuscript by Joshi et al. reports on the GWAS and SNPs for the genes associated with the variation in amino acids and total protein contents in the watermelon seeds. It also provides information on genetic resources that would be useful for breeding targeting the improvement of watermelon seeds for dietary purposes. The significant SNPs and candidate gene information explored in this study would be fundamental for understanding molecular networks controlling variations in the accumulation of specific amino acids and proteins.

Although the manuscript was written excellently and appropriate experiment procedures and analysis were demonstrated, I would like the authors to check the following recommendation to improve their paper.

The introduction is breaken into so many paragraphs and they need to be merged into maximum of 4 -5 paragraphs. The sentence from L71 to L74 may be moved to the next to L62 . The sentence in L78-79 seems not complete.

Please, start L122 as the second paragraph in M&M

There many errors in writing with uniformity for IS unit, Table and Figure in the text, such as L133, L134, L185, L195L237 (figure, Fig, Figure?). L253, Table S3 must be corrected to Supplemental Table 3.

The paragraph extending from L252 to L314 ('Genes under association' section) needs to be broken into several paragraphs.

In Discussion, sentences need to be rearranged accordingly to their contexts. For examples, L323-330 needs to be moved to the next to the sentence on L342

Finally, some information for other approaches and their results for improving protein contents in other crops need to be provided in the Discussion.

---

## Round 0.2 · accepted · Accept

Thank you for your revisions. The revised manuscript is acceptable for publication.

·

Basic reporting

The author have revised the manuscript seriously. Thank you for the author efforts.

Experimental design

No comment

Validity of the findings

No comment

Additional comments

No comment